# Global Spread and Molecular Characterization of CTX-M-Producing *Salmonella* Typhimurium Isolates

**DOI:** 10.3390/antibiotics10111417

**Published:** 2021-11-19

**Authors:** Lili Guo, Yongda Zhao

**Affiliations:** 1Laboratory of Veterinary Pharmacology, College of Veterinary Medicine, Qingdao Agricultural University, Qingdao 266000, China; guolili_house@126.com; 2Laboratory of Veterinary Pharmacology, College of Veterinary Medicine, South China Agricultural University, Guangzhou 510642, China

**Keywords:** CTX-M, *Salmonella* Typhimurium, global distribution, MLST, ARGs

## Abstract

This study aimed to determine the global prevalence and molecular characterization of CTX-M-producing *Salmonella* Typhimurium isolates. A total of 330 (15.2%, 330/21779) *bla*_CTX-M_-positive *S*. Typhimurium were obtained from the public databases in July 2021. Thirteen variants were found in the 330 members of the *bla*_CTX-M_ group, and *bla*_CTX-M-9_ (26.4%, 88/330) was the most prevalent. The majority of *bla*_CTX-M_-positive *S.* Typhimurium were obtained from humans (59.7%, 197/330) and animals (31.5%, 104/330). The number of *bla*_CTX-M_-positive *S.* Typhimurium increased annually (*p* < 0.0001). These isolates were primarily found from China, the United Kingdom, Australia, the USA, and Germany. In addition, these isolates possessed 14 distinct sequence types (ST), and three predominated: ST34 (42.7%, 141/330), ST19 (37.0%, 122/330), and ST313 (10.3%, 34/330). The majority of ST34 *S.* Typhimurium isolates were distributed in China and mainly from swine. However, the majority of ST19 were distributed in the United Kingdom and Australia. Analysis of contigs showed that the major type of *bla*_CTX-M_-carrying plasmid was identified as IncI plasmid (52.9%, 27/51) and IncHI2 plasmid (17.6%, 9/51) in 51 *bla*_CTX-M_-positive S. Typhimurium isolates. In addition, WGS analysis further revealed that *bla*_CTX-M_ co-existed with nine antibiotic-resistant genes with a detection rate over 50%, conferring resistance to five classes of antimicrobials. The 154 virulence genes were detected among these isolates, of which 107 virulence genes were highly common. This study revealed that China has been severely contaminated by *bla*_CTX-M_-positive *S.* Typhimurium isolates, these isolates possessed numerous ARGs and virulence genes, and highlighted that continued vigilance for *bla*_CTX-M_-positive *S.* Typhimurium in animals and humans is urgently needed.

## 1. Introduction

*Salmonella* is an important foodborne pathogen that has a nearly ubiquitous distribution among humans, animals, and the open environment. According to the WHO, there over 90 million people are infected by *Salmonella* annually, and 150,000 people die from *Salmonella* infection [1]. Over 2600 *Salmonella* serotypes have been identified and can be referred to as typhoidal or non-typhoidal (NTS); the latter are present in different animal reservoirs and are responsible for self-limiting gastrointestinal syndromes [2]. *S.* Typhimurium can cause potential threat to human health through the consumption of contaminated food or water [3]. Through estimates of the global invasive non-typhoidal *Salmonella* disease, a study recently found that *S*. Typhimurium is emerging in several African countries [4]. After its oral acquisition, *S.* Typhimurium travels down the intestinal tract and reaches the large intestine, where most of its replication is thought to occur [5].

Treatment of invasive salmonellosis has been compromised due to the emergence of *Salmonella* isolates with single or multidrug resistance to a number of first-line agents. Consequently, the third generation cephalosporins, such as ceftriaxone, have become treatment modalities of choice for therapy against severe *Salmonella* infections [6]. However, the increasing prevalence of cephalosporin-resistant *Salmonella* is also an emerging problem in recent years [7]. Resistance to cephalosporins is mainly due to the acquisition of extended-spectrum β-lactamase genes (ESBLs), among which CTX-M-type enzymes are currently most common—usually located on transmissible plasmids— and can disseminate among the Enterobacteriaceae family [8]. The number of reported cases in various ESBL-producing *Salmonella* serotypes has been increasing worldwide in recent years, with the predominant CTX-M group that was recently detected in poultry and poultry products in different countries [9,10].

In general, *Salmonella* virulence factors have a crucial role in systemic infections [11]. Virulence genes are located on various locations of the genome, including *Salmonella* pathogenicity islands (SPIs) and several mobile genetic elements such as prophages and plasmids [12]. SPI-1 and SPI-2 encode for Type III secretion system (TTSS) and play an important role in delivering different effector proteins into the host that mediates pathogenesis [13]. SPI-1 encodes for genes such as *invA*, *sipABD*, and *sopBD* involved in the invasion of epithelial cells, whereas SPI-2 encodes the genes *sifA* and *ssaR* related to the survival and replication of *Salmonella* within phagocytic cells [14]. In addition, the virulence plasmids carrying virulence genes, such as the *spv* genes, play a role in *Salmonella* multiplication within its host cell, increasing the severity of enteritis [15].

In this study, we first employed genomic analysis from a public database to conduct a global survey of epidemic characterization of CTX-M-producing *S.* Typhimurium. We then characterized the molecular characteristics, diversity, antibiotic-resistant genes, and virulence genotypes of these pathogens.

## 2. Results

### 2.1. Prevalence of Bla_ctx-M_-Producing S. Typhimurium Identified from the Genome Database

In this study, we identified 330 *bla*_CTX-M_-positive *S.* Typhimurium from 21,779 assembled genomes of *S.* Typhimurium from the NCBI database. Thirteen variants were found in the 330 members of the *bla*_CTX-M_ group, of which, the most predominant variants of *bla*_CTX-M_ were *bla*_CTX-M-9_ (26.4%, 88/330), followed by *bla*_CTX-M-55_ (68%, 204/330), *bla*_CTX-M-14_ (19.8%, 66/330), *bla*_CTX-M-15_ (16.5%, 55/330), *bla*_CTX-M-65_ (6.9%, 23/330), and *bla*_CTX-M-1_ (5.7%, 19/330) (Figure 1A). The majority of *bla*_CTX-M_ -positive *S.* Typhimurium were obtained from humans (59.7%, 197/330) and animals (31.5%, 104/330). A noteworthy observation was that swine (18.5%, 61/330) were the primary animal host in (Figure 1B). In addition, the *bla*_CTX-M-9_, *bla*_CTX-M-55_, and *bla*_CTX-M-14_ positive *Salmonella* isolates were obtained from animals and humans, and the latter was more dominant (Appendix A). During this period, we noticed a shift in the quantity of *bla*_CTX-M_-positive *S.* Typhimurium. The percentage of *bla*_CTX-M_-positive *S.* Typhimurium isolates increased from 2.1% (7/330) in 2014 and 10.3% (34/330) in 2015 to 14.2% (47/330) in 2019 and 37.0% (122/330) in 2020 (Figure 1C). The χ^2^ test revealed a significant linear trend among the ordered years from 2014 to 2020 (*p* < 0.0001). These *bla*_CTX-M_-positive *S.* Typhimurium isolates were distributed across 15 countries. China (34.5%, 114/330) has been severely contaminated by *bla*_CTX-M_-positive *S.* Typhimurium isolates, followed by the United Kingdom (28.2%, 93/330), Australia (13.0%, 43/330), USA (5.5%, 18/330), Germany (4.5%, 15/330), and other countries (7.6%, 25/330) (Figure 1D).

### 2.2. Molecular Characterization of Bla_ctx-M_-Producing S. Typhimurium

This group of 330 *bla*_CTX-M_-positive *S.* Typhimurium isolates possessed 14 distinct ST. Three were predominant: ST34 (42.7%, 141/330), ST19 (37.0%, 122/330), and ST313 (10.3%, 34/330), but we could not find a matching ST for 13 of the database isolates (Appendix A). It is worth mentioning that the majority of *bla*_CTX-M_-positive ST34 *S.* Typhimurium isolates were distributed in China (*n* = 94). However, the majority of ST19 were distributed in the United Kingdom (*n* = 63) and Australia (*n* = 34) (Figure 2A). In addition, swine were one of the most dominant hosts in the *bla*_CTX-M_-positive ST34 *S.* Typhimurium isolates (Figure 2B).

### 2.3. Plasmid Analysis

The contigs carrying *bla*_CTX-M_ were confirmed as numerous types of plasmids in 51 *bla*_CTX-M_-positive *S*. Typhimurium isolates. As shown in Figure 3, the major type of *bla*_CTX-M_-carrying plasmid was identified as IncI plasmid (52.9%, 27/51), followed by IncHI2 plasmid (17.6%, 9/51), IncA/C plasmid (7.8%, 4/51), and IncB//O/K/Z plasmid (7.8%, 4/51). Additionally, an analysis of WGS indicated that *bla*_CTX-M_-positive *S.* Typhimurium isolates in this study were carrying 17 types of Inc plasmids (Figure 4). IncHI2 (69.1%, 228/330) and IncHI2A (68.8%, 227/330) were the most prevalent type of plasmid, followed by IncFIB plasmid (33.9%, 112/330) and IncFII plasmid (30.9%, 102/330).

### 2.4. Other ARGs

In addition to bla_CTX-M_, a total of 26 ARGs were identified among these *S*. Typhimurium isolates and conferred resistance to ten classes of antibiotics (Figure 4). Of these, nine ARGs were common with a detection rate of over 50%, including aminoglycoside-resistant genes acc (100%, 330/330), aadA (51.8%, 171/330), ant (63.3%, 209/330), aph (68.8%, 227/330); β-lactam-resistant genes bla_CMY_ (100%, 330/330) and bla_TEM_ (57.9%, 191/330); trimethoprim-resistant gene drfA (60.0%, 198/330); sulphonamide-resistant gene sul (89.7%, 296/330); and tetracycline-resistant gene tet (87.6%, 289/330). Notably, colistin-resistant genes mcr-1, mcr-3, and mcr-9 were detected in 62, 7, and 86 bla_CTX-M_-positive *S.* Typhimurium isolates, respectively. Carbapenem-resistant gene bla_NDM-5_ was detected in two bla_CTX-M_-positive *S.* Typhimurium isolates.

### 2.5. Virulence Factor

Among the 330 *bla*_CTX-M_-positive *S.* Typhimurium isolates, we detected 154 virulence genes (Appendix A). Of these, 107 virulence genes were highly common, with a detection rate over 79%. In contrast, the detection rate of 38 virulence genes were low (<10%). Of note was the detection rate of some virulence genes that ranged from 26.4% to 31.8%, including *grvA*, *spvB*, *spvC*, *spvR*, *pefA*, *pefB*, *pefC*, *pefD*, and *rck* (Figure 5A). The majority of *bla*_CTX-M_-positive *S.* Typhimurium isolates carried 107 virulence genes (34.8%, 115/330), followed by 106 (18.5%, 61/330), 114 (8.8%, 29/330), 115 (8.5%, 28/330), 108 (7.6%, 25/330), and 116 (7.3%, 24/330) (Figure 5B).

## 3. Discussion

At present, it is concerning that the increasing incidence of infections is caused by ESBL-producing organisms, especially *Salmonella* spp., because they are resistant to most of the β-lactam antimicrobials and other antimicrobial classes [16,17]. During the last decade, the most encountered ESBL genes were the CTX-M enzyme family, primarily carried by transferable plasmids and transposons [18]. In this study, all 330 *bla*_CTX-M_-positive *S.* Typhimurium were collected from humans and animals, especially swine. These isolates were distributed across 15 countries. The countries possessing the greatest amount of *bla*_CTX-M_-positive *S.* Typhimurium isolates were China, UK, Australia, USA, and Germany [19,20,21,22,23]. In addition, we noticed an increase in quantity of *bla*_CTX-M_-positive *S.* Typhimurium isolates from 2014 to 2020. This may be related to the increased rate of *Salmonella* after 2009 in food animals [24].

This group of 330 *bla*_CTX-M_-positive *S*. Typhimurium isolates possessed 14 distinct ST, with three predominant: ST34, ST19, and ST313. Previous studies have shown that ST19, ST34, and ST313 are commonly found STs of *S*. Typhimurium. Recently, one of the most prevalent *S*. Typhimurium, ST34, was widely reported in Europe, North America, Asia, and Australia [25]. In addition, ST34 *S*. Typhimurium isolates have acquired strong biofilm-forming ability and were often reportedly carrying the colistin-resistant genes (*mcr-1*, *mcr-3*, *mcr-5*, and *mcr-9*) and even the *bla*_NDM_ gene [25,26,27]. ST19 and ST34 were the most common STs in Asia. However, ST19 isolates were resistant to fewer antibiotic classes than ST34 isolates [27]. Thus far, ST19 has been reported from humans, reptiles, ovine, swine, poultry, food, and bovine from France, Mexico, China, Germany, Scotland, Portugal, Qatar, Korea, Ireland, the United States, the United Kingdom, and Denmark, according to Enterobase [28]. ST313 is a relatively newly emerged sequence type and closely related to the ST19 group of *S.* Typhimurium, which has caused a devastating epidemic of bloodstream infections across sub-Saharan Africa [29].

The *bla*_CTX-M_ genes were mainly located on IncI plasmid in the *S.* Typhimurium isolates, followed by IncHI2 and IncA/C plasmids. Additionally, 17 types of Inc plasmids were identified among the *bla*_CTX-M_-positive *S.* Typhimurium isolates. IncHI2 and IncHI2A were the most prevalent type of plasmid, followed by IncFIB and IncFII plasmids. IncI plasmids were closely associated with the spread of several ESBL genes. The *bla*_CTX-M_ gene was located on IncI1 plasmids in all of the isolates, which were obtained from slaughterhouses located in seven districts in France [30]. A previous study has shown that IncA/C and IncHI2 were the most common types of plasmids among ESBL-producing *Salmonella*, and they often carried multiple antibiotic-resistant genes, including *strB*, *qnrS*, *tet(A)*, *sul*, and *dfrA14* [31]. Additionally, the numerous types of plasmids may serve as important vehicles for the spread of ARGs [32].

In addition to *bla*_CTX-M_, a total of 26 ARGs were identified among the *S.* Typhimurium isolates and conferred resistance to ten classes of antibiotics, including quinolone, aminoglycoside, β-lactam, sulphonamide, trimethoprim, tetracycline, phenicol, macrolide, fosfomycin, and colistin. Consistent with a previous study, most of the *bla*_CTX-M_-positive isolates carried acquired-resistance determinants associated with three or more drug classes [33]. Notably, a few *bla*_CTX-M_ genes co-existed with *mcr-1*, *mcr-3*, *mcr-9*, and *bla*_NDM-5_, conferring resistance to colistin and carbapenem, which were considered as the last-resort antibiotics used for the treatment of infections caused by multidrug-resistant Gram-negative bacteria [34].

An important trait of *Salmonella* is that they can invade, survive, and multiply in a host cell in the presence of genetic determinants for virulence. Virulence genes have been extensively studied in *Salmonella* [35]. In this study, a total of 154 virulence genes were identified in the 330 *bla*_CTX-M_-positive *S.* Typhimurium isolates, and most of the isolates carried 107 virulence genes. Of these, the detection rate of some virulence genes ranged from 26.4% to 31.8%, including *grvA*, *spvB*, *spvC*, *spvR*, *pefA*, *pefB*, *pefC*, *pefD*, and *rck*. The gene *grvA* (for Gifsy-2-related virulence) is located on the prophage Gifsy-2; *grvA* null mutant showed an increase in the virulence of serovar Typhimurium in mice [36]. *SpvB* facilitated *Salmonella* survive and replicate within macrophages via perturbing cellular iron metabolism [37]. The protein encoded by the *spvC* gene has a phosphorylated threonine lyase activity that inhibits MAP phosphokinase. The pathogenicity of *Salmonella* strains greatly increases when both *spvB* and *spvC* genes co-exist in the bacteria [38]. The *spvR* encodes a positive activator for the following four genes, *spvABCD* [39]. Plasmid-encoded fimbriae (Pef) of *Salmonella* are among the fimbriae whose expression have been observed in animals. Pef fimbriae biogenesis depends on the *pef* operon, located on the virulence plasmid of *S*. Typhimurium. This operon encodes the major Pef fimbriae subunit *pefA*, the usher protein *pefC*, required for the assembly of the fimbriae, and the *pefD* periplasmic chaperone for *pefA* [40]. Recent research has shown that *rck* protein is able to induce the *Salmonella* entry mechanism. *rck* mimics natural host cell ligands and triggers engulfment of the bacterium by interacting with the epidermal growth factor receptor [41].

There are several limitations of this study. First, the increase in genomes in the pubic repository may be related to the development of sequencing technology rather than an actual increase. Second, not all genomes of isolates will upload to the database. Finally, the phenotypes of antibiotic resistance and virulence genes and the transfer of resistance genes were not evaluated in this study.

## 4. Materials and Methods

### 4.1. Materials

A total of 21,779 assembled genomes of *Salmonella* Typhimurium were downloaded from the NCBI database in 8 July 2021 https://www.ncbi.nlm.nih.gov/assembly. The detailed information (collected year, hosts, and location) was obtained from the pathogens database (https://www.ncbi.nlm.nih.gov/pathogens, accessed on 8 July 2021).

### 4.2. Methods

All the genomes of *S.* Typhimurium were applied to a filter for the presence of *bla*_CTX-M_ and other antibiotic-resistant genes using ResFinder (https://cge.cbs.dtu.dk/services/ResFinder/ accessed on 29 July 2021). The contigs carrying *bla*_CT*X*-M_ genes were extracted for the identification of plasmid types by PlasmidFinder software. Multilocus sequence types (MLST) and replicating type of plasmid were identified using MLST and PlasmidFinder (https://cge.cbs.dtu.dk/services accessed on 29 July 2021). Virulence factors were identified using the Virulence Factor Database (http://www.mgc.ac.cn/VFs/main.htm accessed on 13 September 2021). The heatmaps were visualized with the “pheatmap” package. The histogram, stack bar diagram, pie diagram, and line chart were plotted with “ggplot2” and colour was used with the “RColourBrewer” package. The χ^2^ test and Fisher’s exact test were used to perform the statistical analysis. For all models, we considered *p* < 0.01 statistically significant and then performed 2-sided probability on those results by using SPSS version 23.0.

## 5. Conclusions

In conclusion, we identified 330 *bla*_CTX-M_-positive *S.* Typhimurium isolates from the public database. Thirteen variants were found in the 330 members of the *bla*_CTX-M_ group, and *bla*_CTX-M-9_ was the most prevalent. The majority of *bla*_CTX-M_-positive *S.* Typhimurium were obtained from humans and animals. The number of *bla*_CTX-M_-positive *S.* Typhimurium increased annually. These isolates were primarily detected in China, the United Kingdom, Australia, the USA and Germany. In addition, these isolates possessed 14 distinct ST, and three were predominant: ST34, ST19, and ST313. The majority of ST34 *S.* Typhimurium isolates were distributed in China and were mainly from swine. However, the majority of ST19 were distributed in the United Kingdom and Australia. The major types of *bla*_CTX-M_-carrying plasmid were identified as IncI plasmid and IncHI2 plasmid. In addition, *bla*_CTX-M_ co-existed with nine antibiotic-resistant genes with a detection rate of over 50%, conferring resistance to five classes of antimicrobials. The 154 virulence genes were detected among these isolates, of which 107 virulence genes were highly common. This study provides new insights into clinical antibiotic therapy for *Salmonella* infection and provides a foundation for further scientific research.

## Figures and Tables

**Figure 1 antibiotics-10-01417-f001:**
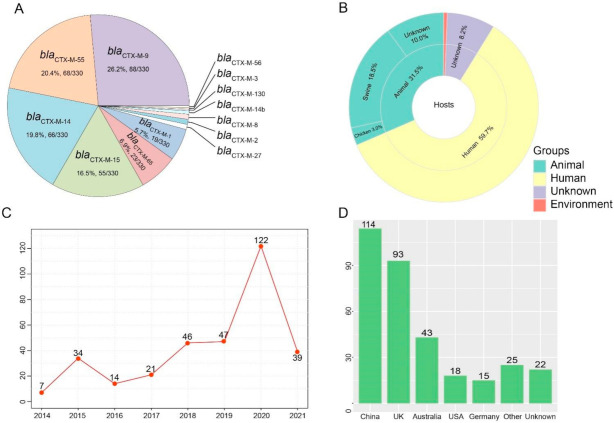
Prevalence of *bla*_CTX-M_-positive *Salmonella* Typhimurium isolates. (**A**) The rate and number of variants in *bla*_CTX-M_ genes. (**B**) The hosts of *bla*_CTX-M_-positive *S.* Typhimurium isolates. (**C**) The number of *bla*_CTX-M_-positive *S.* Typhimurium isolates from 2014 to 2021. (**D**) The number of *bla*_CTX-M_-positive *S.* Typhimurium isolates in different countries.

**Figure 2 antibiotics-10-01417-f002:**
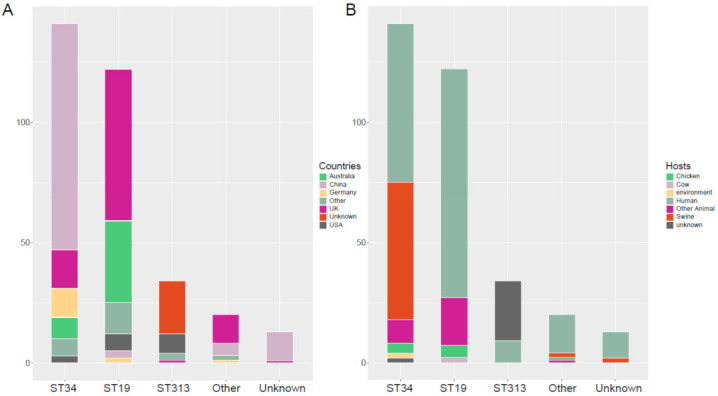
Geographic distribution and host diversity of the dominant ST *bla*_CTX-M_-positive *S.* Typhimurium isolates. (**A**) Geographic distribution the dominant ST *bla*_CTX-M_-positive *S.* Typhimurium isolates. (**B**) Host diversity of the dominant ST *bla*_CTX-M_-positive *S.* Typhimurium isolates.

**Figure 3 antibiotics-10-01417-f003:**
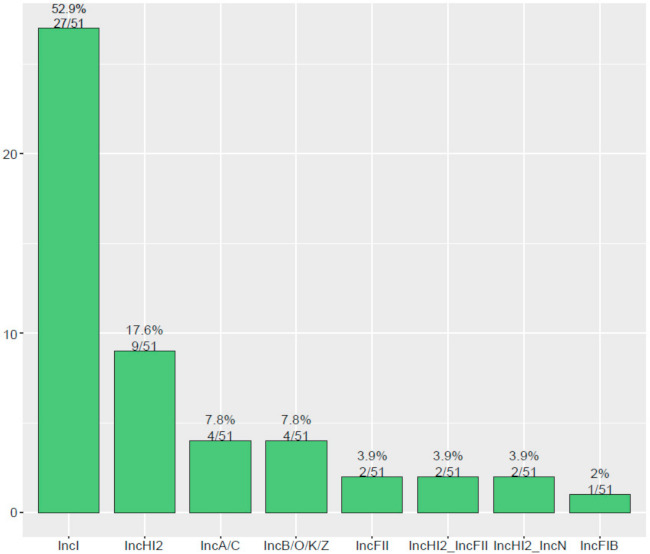
The diversity of plasmid types carrying *bla*_CTX-M_ among 51 *bla*_CTX-M_-positive *S.* Typhimurium isolates.

**Figure 4 antibiotics-10-01417-f004:**
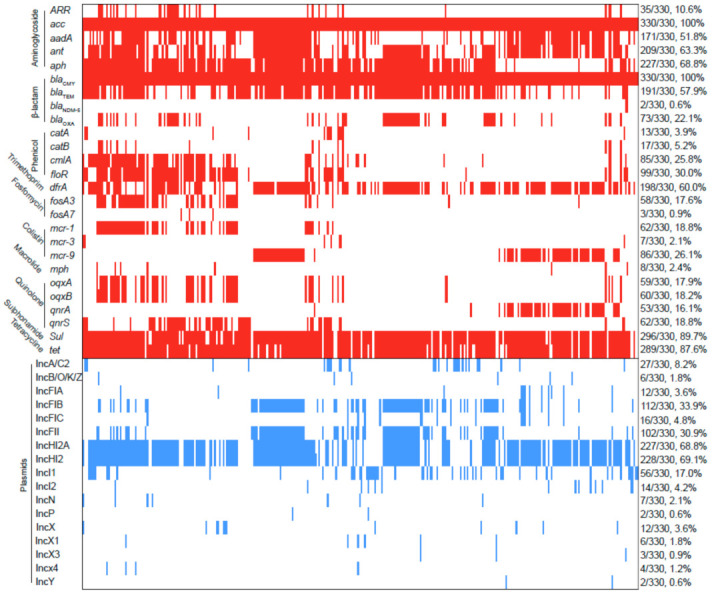
Analysis of ARGs and plasmids among 330 *bla*_CTX-M_-positive *S.* Typhimurium isolates. The red and blue squares represent positivity for ARGs and plasmid Inc types, respectively.

**Figure 5 antibiotics-10-01417-f005:**
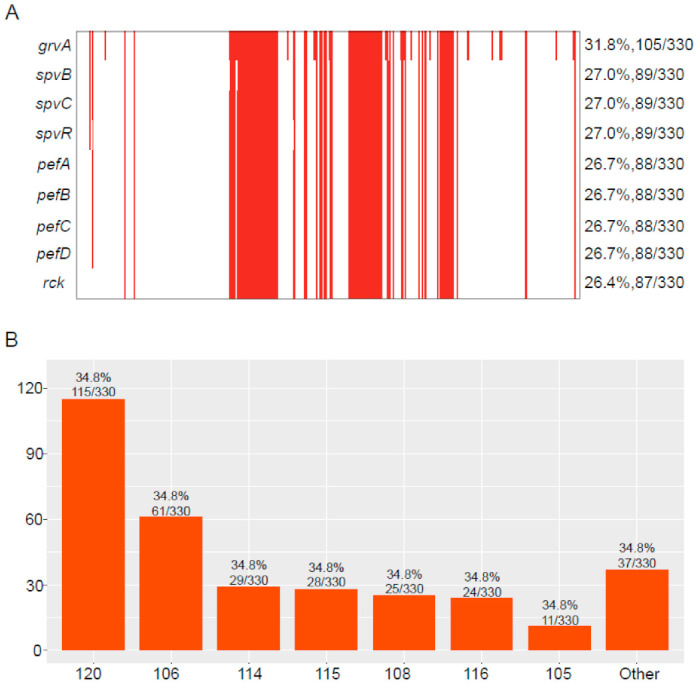
Virulence genes were identified in *bla*_CTX-M_-positive *S.* Typhimurium isolates. (**A**) The detection rate of virulence genes in *bla*_CTX-M_-positive *S.* Typhimurium isolates; the red represents positive for virulence genes. (**B**) The number of isolates carrying different numbers of virulence genes.

## Data Availability

The data presented in this study are available in Appendix A.

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
