# Peer review of "Global Spread and Molecular Characterization of CTX-M-Producing Salmonella Typhimurium Isolates"

_antibiotics, 2021, doi:10.3390/antibiotics10111417_

Round 1

Reviewer 1 Report

  • The major point: whether the research topic is appropriate for the journal “Antibiotics”?
  • The novelty of the study should be underlined in the introduction more clearly
  • The lists of abbreviation used in the text added into the article would make the article more readable
  • The methods are described too enigmatically. I suggest a thorough redrafting and supplementation of this chapter
  • highlighting the eventually clinical benefits of research should be added into the discussion and conclusions

Author Response

Reviewer 1:

The major point: whether the research topic is appropriate for the journal “Antibiotics”?

Response: Thank you for your careful review. This study aimed to investigated the precalence of antibiotics resistance gene (blaCTX-M) in Salmonella Typhimurium, therefore, it is appropriate for the journal “Antibiotics”

The novelty of the study should be underlined in the introduction more clearly

Response: Thank you for your suggestion. We first employed genomic analysis from the database to conduct a global survey of epidemic characterization of CTX-M-producing S. Typhimurium. We added and emphasized the novelty of this study in Line 66-67.

The lists of abbreviation used in the text added into the article would make the article more readable

Response: Thank you for your comments. As showen in Table A2, we added the list of abbreviation used in the text in Line 246.

The methods are described too enigmatically. I suggest a thorough redrafting and supplementation of this chapter

Response: Thank you for your suggestion. We have added the chapter of methods in Line 214-222. The contigs carrying blaCTX-M genes were extracted for the identifition of plasmid types by PlasmidFinder software. The heatmaps were visualized with the “pheatmap” package, the histogram, stack bar diagram, pie diagram, and line chart were plotted with by “ggplot2”, and the colour was uesed with “RColourBrewer” packages. The χ2 test and Fisher’s exact test were used to perform the statistical analysis. For all models, we considered p < 0.01 statistically significant and then performed 2-sided probability on those results by using SPSS version 23.0.

highlighting the eventually clinical benefits of research should be added into the discussion and conclusions

Response: Thank you for your suggestion. We added the eventually clinical benefits in conclusion in Line 235-236. This study provides new insights into clinical antibiotic therapy for salmonella infection, and for laying the foundation for further scientific research.

Reviewer 2 Report

This study focused on the evaluation of the world prevalence and molecular characterization of CTX-M producing Salmonella Typhimurium isolates. Thirteen variants were identified in the 330 isolates mainly in China, United Kingdom, Australia, United States and Germany. Contigs analysis indicated that blaCTX-M plasmids are mainly IncI and IncHI2. In addition, the WGS analysis further demonstrated that blaCTX-M exhibited nine antibiotic resistance genes, conferring resistance to five classes of antimicrobials. These data are very interesting in the context of clinical surveillance, so I suggest accepting the manuscript without any modification.

Author Response

Reviewer 2:

This study focused on the evaluation of the world prevalence and molecular characterization of CTX-M producing Salmonella Typhimurium isolates. Thirteen variants were identified in the 330 isolates mainly in China, United Kingdom, Australia, United States and Germany. Contigs analysis indicated that blaCTX-M plasmids are mainly IncI and IncHI2. In addition, the WGS analysis further demonstrated that blaCTX-M exhibited nine antibiotic resistance genes, conferring resistance to five classes of antimicrobials. These data are very interesting in the context of clinical surveillance, so I suggest accepting the manuscript without any modification.

Response: Thank you very much for your recognition

Reviewer 3 Report

This report provided useful information about the global prevalence and molecular epidemiology of CTX-M-producing Salmonella Typhimurium.

I think the importance of CTX-M-producing Salmonella Typhimurium needs to be describe. Why did you select S Typhimurium among many other nontyphoidal Salmonella diseases?

For example, “Because estimates of the global invasive nontyphoidal Salmonella disease burden have been recently updated and S. Typhimurium is emerging in several African countries(Ref Curr Opin Infect Dis. 2017 Oct 1; 30(5): 498–503).”

Author Response

Reviewer 3

This report provided useful information about the global prevalence and molecular epidemiology of CTX-M-producing Salmonella Typhimurium.

I think the importance of CTX-M-producing Salmonella Typhimurium needs to be describe. Why did you select S Typhimurium among many other nontyphoidal Salmonella diseases? For example, “Because estimates of the global invasive nontyphoidal Salmonella disease burden have been recently updated and S. Typhimurium is emerging in several African countries (Ref Curr Opin Infect Dis. 2017 Oct 1; 30(5): 498–503).”

Response: Thank you for your comments. We cited this article(Line 260) to explained the reason of selected the S. Typhimurium among many other nontyphoidal Salmonella diseases in Line 42-44.

Reviewer 4 Report

The Authors wrote a manuscript on the global spread and molecular characterization of CTX-M-producing Salmonella Typhimurium isolates. Bioinformatics analysis was conducted on S. Thyphimurium genomes deposited in public database.

Major concerns

- The manuscript should be readdressed as review as no experimental data has been carried out, but only a reanalysis of existing data.

 - Since WGS data deposited in online public databases has been carried out starting from different epidemiological settings with different selection criteria, the study cannot estimate a real prevalence of CTX-M-producing S. Typimurium. This limitation should be clearly stated in the text.

-The English language should be extensively revised by a native language medical writer.

Minor concerns

- Figure 1 A and B could be combined to show the hosts carrying most common CTX-M allelic variants.

Author Response

Reviewer 4:

The Authors wrote a manuscript on the global spread and molecular characterization of CTX-M-producing Salmonella Typhimurium isolates. Bioinformatics analysis was conducted on S. Thyphimurium genomes deposited in public database.

Major concerns

The manuscript should be readdressed as review as no experimental data has been carried out, but only a reanalysis of existing data. Since WGS data deposited in online public databases has been carried out starting from different epidemiological settings with different selection criteria, the study cannot estimate a real prevalence of CTX-M-producing S. Typimurium. This limitation should be clearly stated in the text.

Response: Thank you for your comments. We have to admit that there are several limitations of this study, which were added clearly atated in Line 200-204. “Firstly, the increasing of genomes in the pubic repository may be related to the development of sequencing technology, rather than an actual increase. Secondly, not all genomes of isolates will upload the database. Finally, the phenotype of antibiotics resistance and virulence genes, the transfer of resistance genes were not evaluated in this study.”

The English language should be extensively revised by a native language medical writer.

Response: We appreciated this criticism and have the language and grammar of the manuscript checked by a native speaker.

Minor concerns

Figure 1 A and B could be combined to show the hosts carrying most common CTX-M allelic variants.

Response: Thank you for your suggestion. We added a Figure A1 (Line 247) to analysis the hosts carrying most common CTX-M allelic variants, and found that the blaCTX-M-9, blaCTX-M-5 and blaCTX-M-14 positive Salmonella isolates were obtained from animals and human, and the latter was more dominant. We added the result of this analysis in Line 78-80.

Round 2

Reviewer 4 Report

The Authors correctly addressed Reviewers comments.